# An Experimental Study on Enhancing Microbial Desulfurization of Sulfide Ores Using Ultrasonic Treatment

**DOI:** 10.3390/ma15072620

**Published:** 2022-04-02

**Authors:** Wei Pan, Ruge Yi, Zhigang Liao, Lingrong Yang

**Affiliations:** 1School of Resources and Safety Engineering, Central South University, Changsha 410083, China; braveyiruge@163.com (R.Y.); liaozhigang2539@163.com (Z.L.); yanglingrong@126.com (L.Y.); 2Key Laboratory of Ministry of Education of China for Efficient Mining and Safety of Metal Mines, University of Science and Technology Beijing, Beijing 100083, China

**Keywords:** sulfide ore, microbial desulfurization, ultrasonic treatment

## Abstract

Ultrasonic technology is being increasingly explored in minerals processing. In this paper, ultrasonic treatment was introduced as a novel method for microbial desulfurization of sulfide ores. A Box–Behnken experiment was performed to find the best combination of factor levels for the following experiments; consequently, the combination of factor levels at the maximum 5-day average desulfurization rate at 20 g of ore was a particle size of 120–140 mesh, a shaker speed of 175 rpm, and a dosage of 111 mL of bacterial solution. Under these conditions, a microbial desulfurization experiment of sulfide ores by ultrasonic treatment was carried out, and the effect of the particle size, the ultrasonic action time, and the ultrasonic power were investigated. Results indicated that the ultrasonic effect was not significant for ore samples with larger particle sizes, and the appropriate increase in ultrasonic action time was beneficial to the improvement of desulfurization rate, but the effect showed a decreasing trend when it exceeded 50 min, and the best desulfurization effect was achieved when the power was 300 W. This study demonstrated that the effect of microbial desulfurization can be greatly enhanced by ultrasonic treatment.

## 1. Introduction

Since China is one of the countries with a large number of mineral resources, it is vital to ensure the normal production and continuous running of minerals. As a highly hazardous material, spontaneous combustion of sulfide ore is one of the most serious forms of disasters faced by the metal mining process, which, in addition to causing the loss of mineral resources, leads to serious damage to the environment [1,2,3,4]. With the increasing depth of minerals mining, in particular, the problem of rising ground temperature is significant, which will lead to a high possibility of spontaneous combustion and fire accidents [5]. According to statistics, approximately 5–10% of nonferrous metal or polymetallic sulfide ore, and 20–30% of pyrite in China have spontaneous combustion and fire hazards [6]. The composition of sulfide ores is relatively complex, but the main components that can contribute to spontaneous combustion are pyrite, colloidal pyrite, chalcopyrite, etc. [7,8]. Sulfur in these sulfide minerals is the decisive factor for spontaneous combustion in accordance with the mechanism of oxidation and spontaneous combustion [9,10]. Therefore, there is a growing focus on research into desulfurization to curb spontaneous combustion.

It has been proven that microbial desulfurization could be applied to sulfur-ore fire fighting, with the advantages of simple operation, cheap raw materials, mild reaction conditions, and less environmental pollution, compared with other technologies. Hence, many scholars have shifted their attention to microbial desulfurization technology [11].

This technology is currently used in three main fields: petroleum desulfurization, wastewater desulfurization, and coal desulfurization [12,13,14,15,16,17]. Owing to their success, in the field of metal sulfide ores, microbial desulfurization dates back to 1947, when Colmer et al. [18] found that *Thiobacillus ferrooxidans* could accelerate the oxidation of pyrite in coal. Then, Silverman and Stevens began to utilize *Thiobacillus ferrooxidans* to remove pyrite from coal [19,20]. Recently, some researchers conducted experiments to find a negative correlation between the particle size of sulfide ore and desulfurization [21]. Additionally, studies have reported on the utilization of surfactants to strengthen the microbial desulfurization effect [22]. Although various aspects of microbial desulfurization were studied, its efficiency and rate remain low due to the interaction between microorganisms and a relatively complex process on the ore surface, and there is a certain obstruction when the bacterial solution is in contact with the ore [23].

In recent years, ultrasound has become increasingly popular and widely used as a novel technical tool in mining-related experiments [24,25,26,27,28]. Some studies have found that ultrasound could change the potential of the ore and have a crushing effect on selected ore particles, as well as change the pH and temperature of the pulp. Furthermore, it was found that not only the sulfur and iron elements on the surface of the ore were chemically displaced, but the relative density of anions and cations on the surface also changed to some extent by using an ultrasonic treatment on pyrite. Hence, the study showed that ultrasonic treatment could change the hydrophobicity and hydrophilicity of metal sulfides, in addition to significantly improving floatability [29,30,31]. In addition, ultrasonic treatment could greatly improve the interface area for the reaction, increase the effective local concentration of the reactive species, and enhance the mass transfer in the interface area. Therefore, it led to a significant increase in solubility [32,33]. Regrettably, there is still a gap in research related to the combination of ultrasound and microorganisms to remove sulfur from sulfide ores.

Therefore, using ultrasonic treatment to enhance the techniques for microbial desulfurization is an important research direction. Based on the above considerations, an experimental scheme was designed with ore samples collected from a pyrite mine in China as the experimental materials. Firstly, a steepest-climb experiment was conducted to quickly reach the response surface center to ensure that the subsequent response surface fitting equation was accurate and valid. Then, a Box–Behnken experiment was performed with a response surface center to find the best factor level combination for the following experiments. Finally, microbial desulfurization experiments of sulfide ores were carried out by ultrasonic treatment, focusing on the effect of particle size, ultrasonic treatment time, and ultrasonic power on desulfurization rate. In summary, this study has important theoretical and practical significance for the sustainability of the minerals.

## 2. Materials and Methods

### 2.1. Sources of Ore Samples

Compared with other types of ores, elemental sulfur in sulfide ores is an important internal factor in the occurrence of spontaneous combustion of the ore, and its content will cause sulfide ores to have a faster oxidation rate. Therefore, to prevent spontaneous combustion fire accidents at the root, we need to focus on the high sulfur content of the ore itself.

The presence of elemental sulfur in sulfide ores is in the form of organic and inorganic sulfur, with inorganic sulfur consisting mainly of sulfides (FeS, FeS_2_) and sulfates (CaSO_4_, BsSO_4_, FeSO_4_). Reducing the occurrence of spontaneous combustion accidents in sulfide ores from the endogenous point of view mainly requires reducing the content of inorganic sulfur (FeS, FeS_2_) in the ore.

From the microbiological point of view, for inorganic sulfur desulfurization mechanism can be roughly divided into the direct action of bacteria, indirect action of bacteria, and compound action of bacteria. By cultivating bacteria with elemental sulfur as the main nutrient and through a series of reactions, the sulfur content of sulfide ore can be fundamentally reduced.

Therefore, in this paper, we selected sulfide ore as the subject. The sulfide ore samples (mainly containing pyrite) were collected from Tongshan, Anhui Province, China. Many cases of accidents due to spontaneous combustion of sulfide ore have been reported in Tongshan, which brought large economic losses and casualties, so the selection of ore samples from this site is beneficial to provide a basis for production practice. Mineral composition analysis shows that the main elements of the sulfur ore samples were sulfur, iron, oxygen, and silicon, with small amounts of calcium, barium, magnesium, and other elements. The main compositions of the sulfide ore samples are listed in Table 1.

### 2.2. Strain and Medium

In the previous experimental studies, the desulfurization rate of *Acidithiobacillus caldus* (*A.c.*) was higher than all other bacteria, so *A.c.* was selected as the primary strain for the experiment. The medium used for *A.c.* was 9K medium. The specific composition of this medium is shown in Figure 1. Briefly, 5 g of sulfide mineral powder was added to the medium and incubated at 43 °C for 5 days (the rotating speed of a constant temperature shaker was set at 170 r/min) until the bacteria count reached a certain number. All chemicals were of analytical grade.

### 2.3. Optimization of Experimental Conditions

#### 2.3.1. Design of the Steepest Climb Experiment

Aiming to determine the response surface center, the steepest-climb experiment was designed. All three factors have a positive climbing direction, so their levels should be gradually increased in the experiment. According to the effect value analysis, the following incremental steps were set: (A) particle size of ore sample as 20, (B) the speed of the shaker as 10, and (C) the dosage of bacterial solution as 10. Specifically, as the level value gradually increased by one unit, each of the three factors should increase by the increment indicated in their corresponding steps. The experimental level design of the steepest-climb experiment is shown in Table 2.

Five groups of experimental samples were set up, with two samples in each group. The specific process of the experiment was as follows:(1)A sufficient quantity of ore samples of sizes 60–80 mesh, 80–100 mesh, 100–120 mesh, 120–140 mesh, and 140–160 mesh were taken for the experiment; then, 20 g of ore samples were taken from each bottle according to size and numbered and acidified with pH = 1 sulfuric acid; each size of ore samples was then dipped in sulfuric acid for 1–2 days to bring their surface pH down to 2.5.(2)A quantitative amount of bacterial solution and leaching aid was added to the experimental ore samples, and the leaching aid was diluted to a concentration of 0.01% using the culture medium, with the bacterial solution: leaching aid = 5:1. The initial sulfur concentration test samples were extracted.(3)The mixed ore samples and desulfurization solution were sealed and put into a shaker, with the temperature set at 35 °C. The sulfur concentration test samples were extracted for 5 days.

#### 2.3.2. Design of the Box–Behnken Experiment

Aiming to find the best factor level combination, a Box–Behnken experiment was designed with three factors and three levels. The upper, lower, and zero levels were set for three factors, where the zero level was set based on the response surface center, +1 means upper level, and −1 means lower level. The experimental level design of the Box–Behnken experiment is shown in Table 3.

In total, 15 groups of experimental samples were set up, and the specific procedure was as follows:(1)A sufficient quantity of ore samples of 100–120 mesh, 120–140 mesh, and 140–160 mesh were taken for the experiment; then, 20 g of each sample was taken from each bottle of the three numbered sizes and acidified with pH = 1 sulfuric acid. Each particle size of the samples was then dipped in sulfuric acid for 1–2 days to bring the samples’ surface pH down to 2.5.(2)A quantitative amount of bacterial solution and leaching aid was added to the experimental ore samples, and the leaching aid was diluted to a concentration of 0.01% using the culture medium, with the bacterial solution: leaching aid = 5:1. The initial sulfur concentration test samples were extracted.(3)The mixed ore samples and desulfurization solution were sealed and put into a shaker, with the temperature set at 35 °C. The sulfur concentration test samples were extracted for 5 days. All 15 sets of experimental samples are shown in Figure 2.

### 2.4. Microbial Desulfurization Experiments of Sulfide Ores Using Ultrasonic Treatment

In this experiment, we explored the influence of ultrasonic treatment on the effect of microbial desulfurization. The ultrasonic device is shown in Figure 3. Meanwhile, the control variable method was adopted to study the main factors affecting microbial desulfurization by ultrasonic treatment, which include the particle size of ore samples, the ultrasonic action time, and the ultrasonic power. The sulfur content in sulfide ore and the desulfurization rate of microorganisms were analyzed under different parameter conditions so that we could find the optimal parameter conditions.

The experimental flow and setups of microbial desulfurization experiments of sulfide ores using ultrasonic treatment are shown in Figure 3.

Firstly, the ore sample was mixed with distilled water proportionally to make the ore sample treated more fully. Then, the ore sample reacted under ultrasonic conditions. As the pH in the ore slurry was not suitable for the survival of bacteria, it needed to be acidified before adding bacteria. After acidification, bacterial solution and leach were added proportionally, and it was incubated in a constant temperature shaker for 5 days. Additionally, the supernatant was taken daily to calculate the sulfur content of the solution and desulfurization rate.

#### 2.4.1. Experiment of the Effect of the Particle Size of Ore Sample on Desulfurization Rate

The particle size of the ore sample affected the reaction between the sulfur of the ore and the microorganism. Thus, the specific process of the experiment was as follows:(1)Four different particle sizes of 60–80 mesh, 80–100 mesh, 100–120 mesh, and 120–140 mesh were selected. Then, 20 g of each of the ore samples was added to distilled water to form a slurry for ultrasonic treatment and then filtered dry and acidified to make the pH of the ore samples 2.0.(2)The incubated bacterial solution was mixed with the same amount of 9K medium (diluted at a ratio of 1:10). Combined with the previous research results, leach (diluted to 0.005%) was added to the bacterial solution.(3)After pretreatment with ultrasonic power of 200 W and an action time of 30 min, 111 mL of A.c. was added and incubated in a constant temperature shaker at 35 °C and 175 r/min for 5 days (111 mL of *A.c.* and 175 r/min for shaker were determined by the Box–Behnken experiment).

The design of the experiment on the effect of the particle size of ore sample on desulfurization rate is shown in Table 4.

#### 2.4.2. Experiment of the Effect of the Ultrasonic Action Time on Desulfurization Rate

In this experiment, the best desulfurization effect of particle size 120–140 mesh was selected (according to the results of the experiment on the effect of the particle size of ore sample on desulfurization rate); other conditions remained unchanged, and ultrasonic action time was used as a variable to study the effect of the ultrasonic action time on desulfurization rate. According to the results of the previous pre-experiments, there was an obvious difference in the desulfurization effect when choosing 1 min, 2 min, and 3 min. Hence, the minimum ultrasonic action time was set to 30 min, and the incremental step was 10 min.

The design of the experiment on the effect of the ultrasonic action time on the desulfurization rate is shown in Table 5.

#### 2.4.3. Experiment of the Effect of the Ultrasonic Power on Desulfurization Rate

In this experiment, ultrasonic power was used as a variable to study the effect of ultrasonic power on the desulfurization rate. Ultrasonic power was set to 100 W, 200 W, 300 W, and 400 W. Ultrasonic action time was set to 50 min, but other conditions remained unchanged.

The design of the experiment on the effect of the ultrasonic power on the desulfurization rate is shown in Table 6.

## 3. Results

In this paper, arithmetic averages were used to determine the final desulfurization rate throughout the data analysis and calculations.
(1)$$P_{i}=\frac{\alpha_{i}\;-\;\alpha }{i}\;\times \;V$$

In this formula:
$P_{i}$—The desulfurization rate on day *i*, in mg/d;*i*—Reaction days, in d;$\alpha_{i}$—Sulfur concentration in bacterial solution after *i* days of desulfurization, in mg/L;α—Sulfur concentration in bacterial solution before desulfurization, in mg/L;V—Volume of desulfurization solution, in L.


### 3.1. Results of Optimization of Experimental Conditions

#### 3.1.1. Results of the Steepest-Climb Experiment

After 5 days of the experimental cycle, the actual value of sulfur content of the experimental samples was measured, and the 5-day average desulfurization rate of microorganisms was calculated by Formula (1). The 5-day average desulfurization rate results for each group of samples were averaged to yield the results of the steepest-climb experiment, as shown in Table 7.

From Table 7, it can be seen that the 5-day average desulfurization rate of the experimental samples in group 4 reached the highest value, so X + 4Δx was selected as the response surface center, and the level of the center was set as follows: (A) the particle size of ore sample was 120–140 mesh, (B) the speed of the shaker was 170 rpm, and (C) the dosage of the bacterial solution was 110 mL.

#### 3.1.2. Results of the Box–Behnken Experiment

After 5 days of the experimental cycle, the actual value of sulfur content of the experimental samples was measured, and the 5-day average desulfurization rate of microorganisms was calculated by Formula (1). The 5-day average desulfurization rate results for each group of the sample were averaged to yield the results of the Box–Behnken experiment, as shown in Table 8.

The data were imported into Design-Expert software for model fitting analysis, and the multivariate quadratic regression equation of the obtained response value with the three factors—namely, the particle size of the ore sample, the speed of the shaker, and the dosage of the bacterial solution—is shown in Formula (2).
(2)$$R=-772.624+12.44446A-12.6148B+18.18567C-0.00452\mathrm{AB}-0.01405\mathrm{AC}+0.0212\mathrm{BC}-0.03755A^{2}+0.035783B^{2}-0.09012C^{2}$$
where R is the response value, which is the 5–day average desulfurization rate.

The predicted value of the 5–day average desulfurization rate can be calculated according to Formula (2). Therefore, the correspondence between the actual and predicted values drawn in the coordinate system is shown in Figure 4.

As evident from Figure 4, the predicted value introduced by the equation was close to the actual response value, so the equation was relatively accurate. Additionally, as shown in Figure 5, the correspondence between model residual and predicted value proved the well-fitting of the model, which indicated that it was feasible to optimize the experimental conditions for microbial desulfurization of sulfide ore using the response surface methodology from another perspective.

The significance of the experimental factors and their interactions were analyzed by Design-Expert software, and the results of the analysis of variance for the Box–Behnken experiment are shown in Table 9.

As indicated in the listed results in Table 9, the F value of the response surface model was 406.94, with a *p*-value < 0.0001, which indicated that the entire model fit was significantly reliable and statistically significant.

The experimental results were optimized by constraining the level setting range of the factors and nonlinear programming to find the optimal level combination of three factors. The combination of factor levels at the maximum 5-day average desulfurization rate was as follows: the particle size of the ore sample was 120–140 mesh, the speed of the shaker was 175 rpm, and the dosage of the bacterial solution was 111 mL.

### 3.2. Results of Microbial Desulfurization Experiments of Sulfide Ores by Ultrasonic Treatment

#### 3.2.1. Analysis of the Effect of Particle Size of Ore Sample on Desulfurization Rate

After 5 days of the experimental cycle, the actual value of sulfur content of the experimental samples was measured, and the 5-day average desulfurization rate of microorganisms was calculated by Formula (1). The results are shown in Table 10 and Figure 6.

It can be seen from Table 10 and Figure 6 that the average desulfurization rate increased after ultrasonic treatment for different particle sizes, compared with the raw ore samples. Microbial desulfurization rate was in the following order: 120–140 mesh > 100–120 mesh > 80–100 mesh > 60–80 mesh. The result was consistent with the optimal level of particle size in the Box–Behnken experiment; therefore, it also confirmed the validity of the Box–Behnken experimental results.

As can be seen in Figure 7, microbial desulfurization in each group of ore samples showed an ascending and then descending trend, which was related to the growth cycle of bacteria, with four stages, i.e., retardation, logarithm, stability, and decay.

Before *A.c.* in the solution used the material in the ore samples to grow and multiply, it first needed a process to adapt to the environment, so the effect of microorganisms was not apparent. As the microorganisms’ adaptability increased, they reached the logarithmic growth period, during which they underwent rapid growth and reproduction; at this time, the desulfurization rate reached the peak of the desulfurization cycle and then stabilized for a certain time, which was the stable period of bacterial growth. Due to the limited material and growth space within the ore samples, the bacteria would enter the last stage, i.e., decay. During this period, the desulfurization capacity of bacteria diminished, and the sulfur content of the ore surface decreased. Therefore, when conducting microbial desulfurization experiments, the growth cycle and the optimal duration of desulfurization should be considered in order to achieve the beat desulfurization effect in a short time.

As can be seen in Figure 8, the surface of the untreated ore was relatively smooth, and there was no trace of destruction, while the surface of the ultrasonically treated ore had many large pores, which made it easier for microorganisms to attach to the surface of the ore for reaction. The pores were formed by the cavitation effect of ultrasound, which was a clear explanation for the enhanced desulfurization effect.

#### 3.2.2. Analysis of the Effect of Ultrasonic Action Time on Desulfurization Rate

After 5 days of the experimental cycle, the actual value of sulfur content of the experimental samples was measured, and the 5-day average desulfurization rate of microorganisms was calculated by Formula (1). The results are shown in Table 11 and Figure 9.

It can be seen from Table 11 and Figure 9 that the ultrasonic action time played a significant role in the effect of desulfurization under ultrasonic conditions. When the ultrasonic action time was less than 50 min, the desulfurization rate increased with the improvement in ultrasonic action time, and when the ultrasonic action time exceeded 50 min, the desulfurization rate showed a decreasing trend, so the best desulfurization effect was at the ultrasonic action time of 50 min.

#### 3.2.3. Analysis of the Effect of Ultrasonic Power on Desulfurization Rate

After 5 days of the experimental cycle, the actual value of sulfur content of the experimental samples was measured, and the 5-day average desulfurization rate of microorganisms was calculated by Formula (1). The results are shown in Table 12 and Figure 10.

By analyzing Table 12 and Figure 10, it can be seen that the microbial desulfurization rate increased with the improvement of ultrasonic power, but after 300 W, it showed a decreasing trend, and the average desulfurization rate at the power of 400 W was lower than that at 100 W, but overall, it was better than the desulfurization rate of untreated samples.

The reason for this phenomenon may be related to the mechanism of ultrasonic action. In the beginning, with the increase in ultrasonic power, ultrasonic treatment made the cavitation bubbles produced in the pulp increase; then, the physical and chemical effects of cavitation became more apparent, and the surface pores of the ore sample, as well as the contact areas of bacteria and ore, which are conducive to the reaction between microorganism and ore samples, increased. However, As the ultrasonic power is excessively high, a large number of cavitation bubbles were generated around the ultrasonic amplitude rod, and the bubbles would shield the generation of the effect and hinder the distribution of ultrasonic energy in the entire solution environment, which would lead to a lower conversion efficiency of ultrasonic treatment. Thus, the desulfurization rate was reduced.

#### 3.2.4. Determination of Surface Sulfur Content of Ore Samples by EDS

By analyzing Figure 11, it can be concluded that the sulfur content on the surface of the ultrasonically treated ore samples was lower than that in the untreated ore samples. This remained consistent with the conclusions drawn from the above experiments.

## 4. Conclusions

The aim of this study was to investigate the use of ultrasonic treatment to treat sulfide ores, so as to enhance microbial desulfurization. On the basis of the results obtained, the conclusions can be made as follows:(1)The response surface center was determined by conducting a steepest-climb experiment, and the Box–Behnken experiment was carried out with it. The fit analysis and ANOVA of the experimental result model showed that it was a well-fitted model. Based on the above results, a combination of factor levels at a maximum 5-day average desulfurization rate was found by constraining the level setting range of the factors as follows: the particle size of the ore sample was 120–140 mesh, the speed of the shaker was 175 rpm, and the dosage of the bacterial solution was 111 mL.(2)Through microbial desulfurization experiments of sulfide ores using ultrasonic treatment under an optimal combination of factor levels, it was confirmed that the application of ultrasonic treatment enhanced the microbial desulfurization effect. In addition, it was found that the ultrasonic effect was not obvious for ore samples with larger particle sizes, and the appropriate increase in ultrasonic action time was beneficial to the improvement of the desulfurization rate, but the effect showed a decreasing trend when it exceeded 50 min, and the appropriate increase in ultrasonic power reduced the particle size of the ore samples; the best desulfurization effect was achieved when the power was 300 W.

## Figures and Tables

**Figure 1 materials-15-02620-f001:**
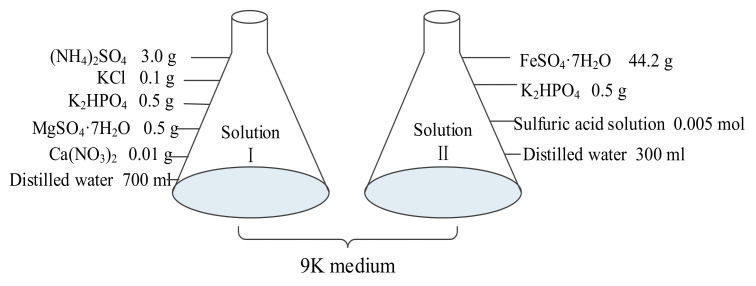
Composition of the 9K medium.

**Figure 2 materials-15-02620-f002:**
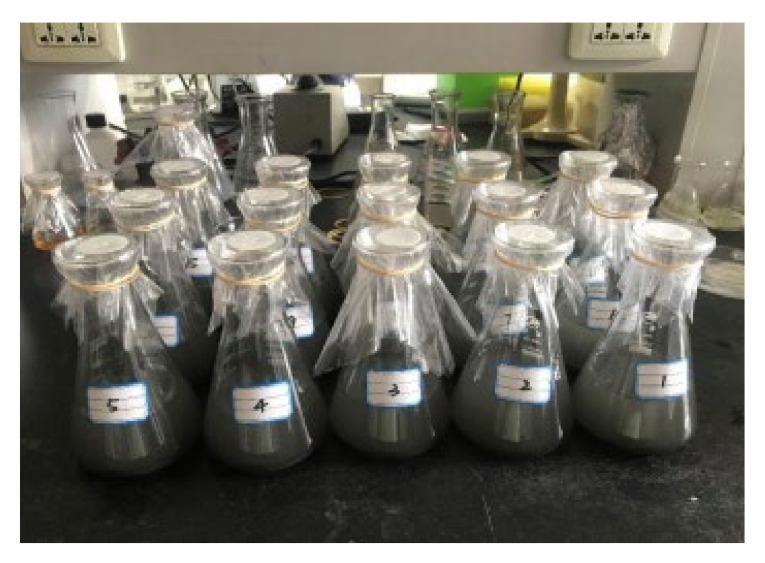
Box–Behnken experimental samples.

**Figure 3 materials-15-02620-f003:**
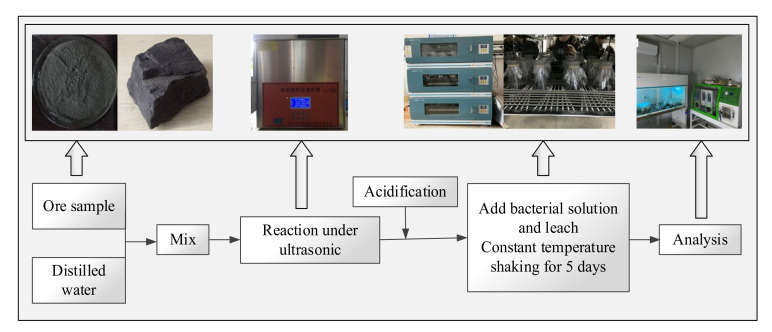
The experimental flow of microbial desulfurization experiments of sulfide ores using ultrasonic treatment.

**Figure 4 materials-15-02620-f004:**
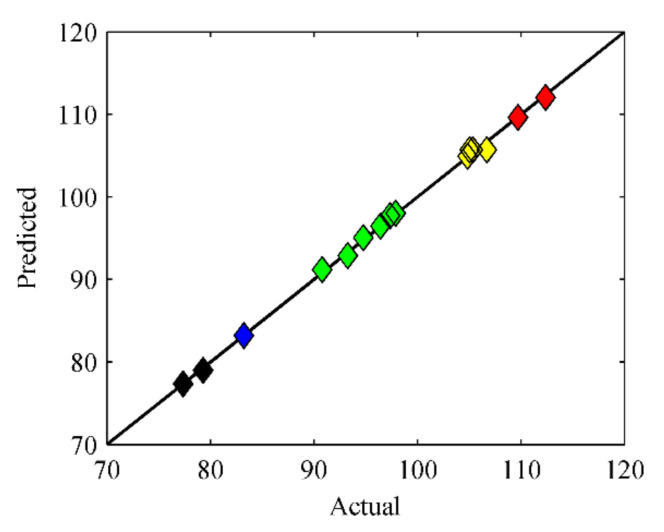
Corresponding relationship between the predicted and actual value of average desulfurization rate (Different colors indicate different ranges of actual value of average desulfurization rate).

**Figure 5 materials-15-02620-f005:**
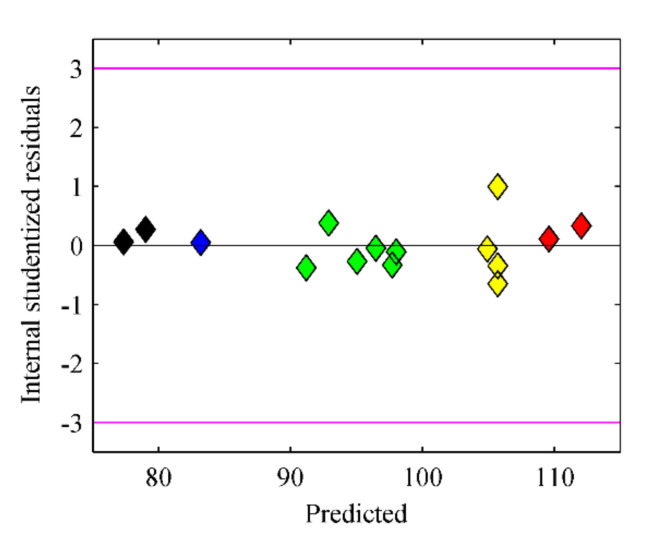
Corresponding relationship between the model residual and predicted value (Different colors indicate different ranges of actual value of average desulfurization rate).

**Figure 6 materials-15-02620-f006:**
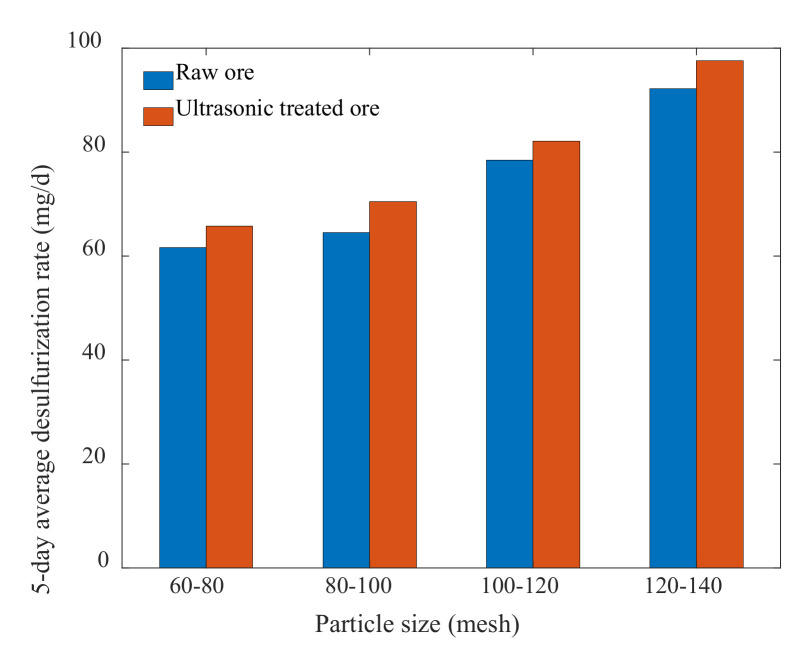
Comparison of the 5–day average desulfurization rates of different particle sizes.

**Figure 7 materials-15-02620-f007:**
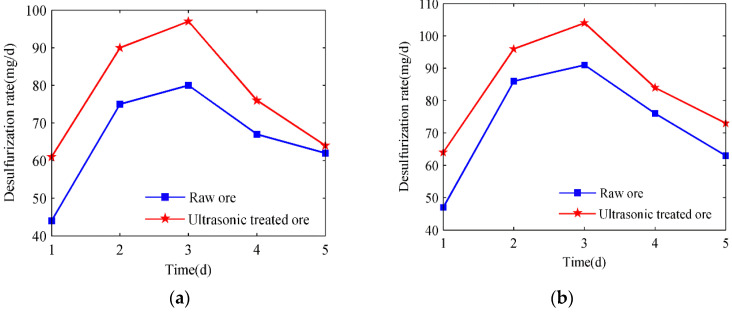
Comparison of the desulfurization rate with time for different particle sizes: (**a**) 60–80 mesh; (**b**) 80–100 mesh; (**c**) 100–120 mesh; (**d**) 120–140 mesh.

**Figure 8 materials-15-02620-f008:**
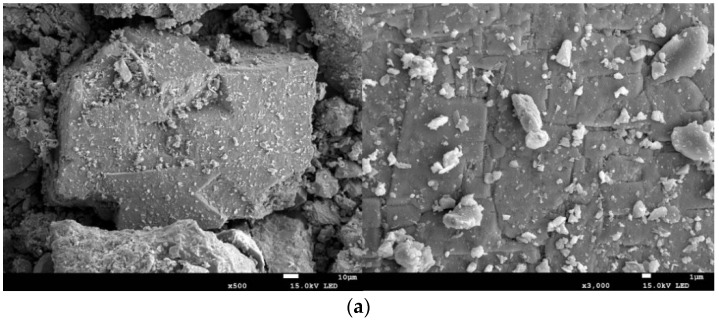
Scanning electron microscopic images of the ore sample before and after ultrasonic treatment: (**a**) untreated ore; (**b**) ultrasonically treated ore.

**Figure 9 materials-15-02620-f009:**
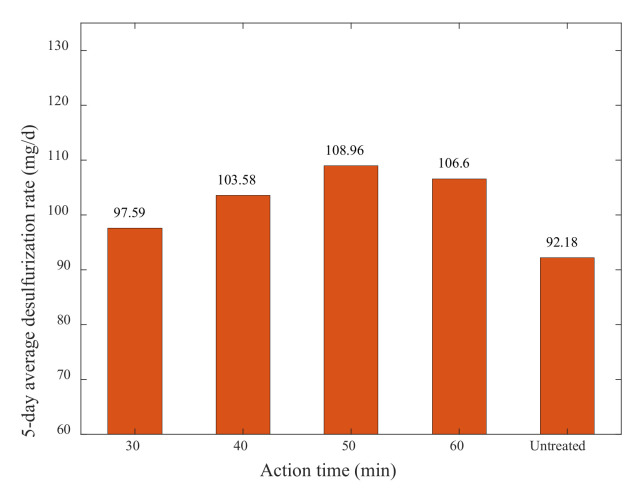
Comparison of the 5–day average desulfurization rates of different ultrasonic action times.

**Figure 10 materials-15-02620-f010:**
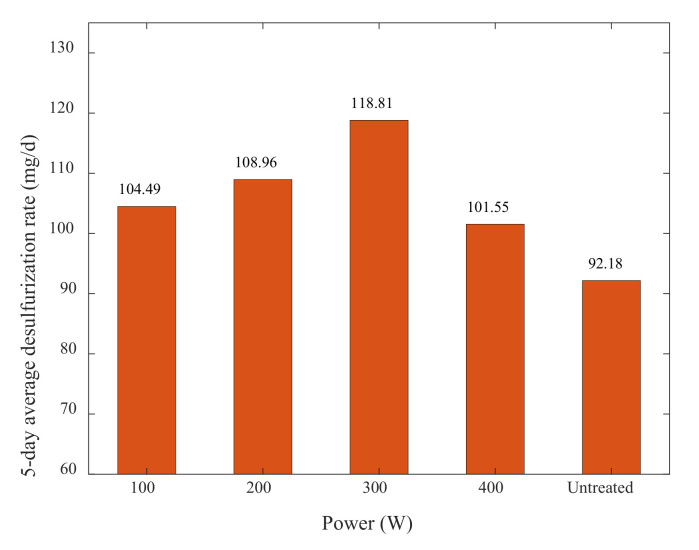
Comparison of the 5-day average desulfurization rate of different ultrasonic power.

**Figure 11 materials-15-02620-f011:**
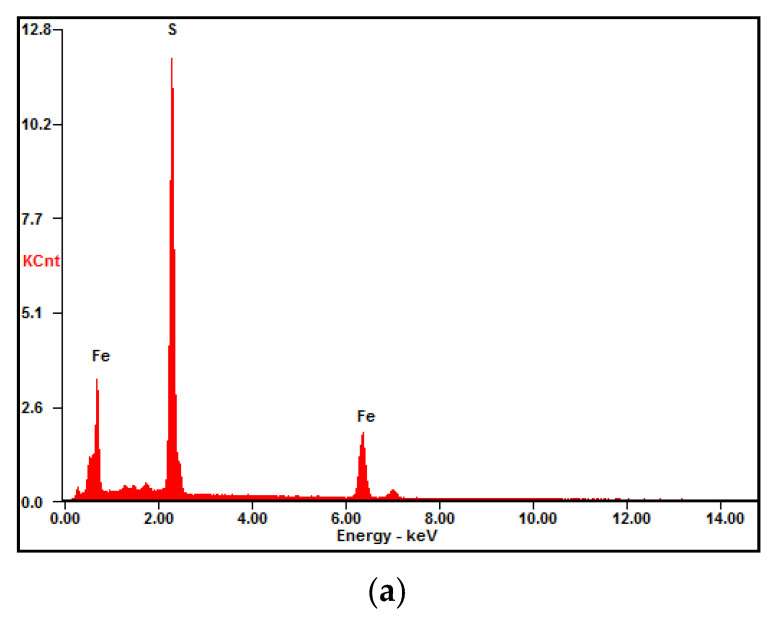
EDS spectrum: (**a**) untreated ore; (**b**) ultrasonically treated ore.

**Table 1 materials-15-02620-t001:** Main chemical composition (wt%) and mass content of the sulfide ore samples.

S	Fe	O	Si	Ca	Ba	Mg	Al
37.33	32.51	11.1	7.265	2.47	1.94	1	0.772
Cu	As	K	Zn	Mn	Na	Ti	Sr
0.307	0.246	0.121	0.115	0.0577	0.049	0.027	0.0258
P	Ag	Pb	Cr	Mo			
0.022	0.013	0.013	0.011	0.004			

**Table 2 materials-15-02620-t002:** Experimental level design for the steepest-climb experiment.

Number	Level Value	A(mesh)	B(rpm)	C(mL)
1	X + Δx	60–80	140	80
2	X + Δ2x	80–100	150	90
3	X + Δ3x	100–120	160	100
4	X + Δ4x	120–140	170	110
5	X + Δ5x	140–160	180	120

**Table 3 materials-15-02620-t003:** Experimental level design for the Box–Behnken experiment.

Number	Factor	Low Level (−1)	Zero Level (0)	Upper Level (+1)
A	Particle size of ore sample	100–120	120–140	140–160
B	Speed of the shaker	165	170	175
C	Dosage of bacterial solution	105	110	115

**Table 4 materials-15-02620-t004:** Experimental design for the effect of the particle size of ore sample on desulfurization rate.

Particle Size (Mesh)	60–80	80–100	100–120	120–140
Category (Experiment number)	Raw ore (Y-1)	Raw ore (Y-2)	Raw ore (Y-3)	Raw ore (Y-4)
	Treated ore (C-1)	Treated ore (C-2)	Treated ore (C-3)	Treated ore (C-4)

**Table 5 materials-15-02620-t005:** Experimental design for the effect of the ultrasonic action time on desulfurization rate.

Time (min)	30	40	50	60
Category (Experiment number)	Treated ore (C-3)	Treated ore (C-4)	Treated ore (C-5)	Treated ore (C-6)

**Table 6 materials-15-02620-t006:** Experimental design for the effect of the ultrasonic power desulfurization rate.

Power (W)	100	200	300	400
Category (Experiment number)	Treated ore (C-1)	Treated ore (C-2)	Treated ore (C-3)	Treated ore (C-4)

**Table 7 materials-15-02620-t007:** Results of the steepest-climb experiment.

Number	Value	A(mesh)	B(rpm)	C(mL)	5-Day Average Desulfurization Rate (mg/d)
1	X + Δx	60–80	140	80	30.73
2	X + Δ2x	80–100	150	90	43.46
3	X + Δ3x	100–120	160	100	64.75
4	X + Δ4x	120–140	170	110	88.70
5	X + Δ5x	140–160	180	120	86.79

**Table 8 materials-15-02620-t008:** Results of the Box–Behnken experiment.

Number	A	B	C	5–Day Average Desulfurization Rate (mg/d)
1	1	0	−1	96.42
2	1	0	1	94.76
3	0	−1	−1	97.37
4	−1	−1	0	77.37
5	0	1	1	112.36
6	0	−1	1	97.91
7	0	1	−1	109.70
8	0	0	0	106.69
9	1	1	0	104.86
10	0	0	0	105.04
11	1	−1	0	93.25
12	−1	0	1	83.23
13	−1	0	−1	79.27
14	−1	1	0	90.79
15	0	0	0	105.35

**Table 9 materials-15-02620-t009:** Analysis of variance for the Box–Behnken experiment.

Source	Sum of Square	Degree of Freedom	Mean Square	F Value	*p* Value
Model	1633.13	9	181.46	406.94	<0.0001
A	429.76	1	429.76	963.78	<0.0001
B	335.60	1	335.60	752.63	<0.0001
C	3.77	1	3.77	8.46	0.0335
AB	0.82	1	0.82	1.84	0.2326
AC	7.88	1	7.88	17.66	0.0085
BC	1.12	1	1.12	2.51	0.1737
A^2^	832.82	1	832.82	1867.69	<0.0001
B^2^	2.95	1	2.95	6.61	0.0500
C^2^	18.73	1	18.73	41.99	0.0013
Residual	2.23	5	0.45	-	-
Lack of fit	0.69	3	0.23	0.30	0.8270
Pure error	1.54	2	0.77	-	-
Cor total	1635.36	14	-	-	-

**Table 10 materials-15-02620-t010:** Results of the experiment on the effect of particle size on desulfurization rate.

Number	Concentration of Elemental Sulfur (mg/L)	
	Initial Concentration	Day 5
Y-1	2219.8	4994.8
C-1	2224.5	5186.5
Y-2	2250.2	5154.8
C-2	2182.1	5355.3
Y-3	2342.8	5875.8
C-3	2375.9	6076.4
Y-4	2350.3	6502.6
C-4	2642.5	7038.5

**Table 11 materials-15-02620-t011:** Results of the experiment on the effect of ultrasonic action time on desulfurization rate.

Number	Concentration of Elemental Sulfur (mg/L)	
	Initial Concentration	Day 5
Raw ore	2350.3	6502.6
C-3	2642.5	7038.5
C-4	2651.8	7317.4
C-5	2583.4	7491.6
C-6	2604.0	7405.6

**Table 12 materials-15-02620-t012:** Results of the experiment on the effect of ultrasonic power on desulfurization rate.

Number	Concentration of Elemental Sulfur (mg/L)	
	Initial Concentration	Day 5
Raw ore	2350.3	6502.6
C-1	2476.8	7183.4
C-2	2583.4	7491.6
C-3	2652.9	8004.9
C-4	2531.2	7150.5

## Data Availability

This study did not report any data.

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
