# Peer review of "An Experimental Study on Enhancing Microbial Desulfurization of Sulfide Ores Using Ultrasonic Treatment"

_materials, 2022, doi:10.3390/ma15072620_

Round 1

Reviewer 1 Report

This interesting paper deals by microbial desulfurization of sulphide ores (mainly containing pyrite) by additional ultrasonic treatment. The research is well-designed by experimentally varying factors like mesh size of the ore, dosage of bacterial solution, shaker speed, duration and power of ultrasonic action. For sure, there is a valuable novelty in ore treatment and thus this is a valuable contribution to the ore treatment procedures, which is recommendable for publishing. Prior to publishing, I would suggest minor changes in the manuscript, essentially in Materials and methods chapter – in some places just a bit more description is required in order to understand the complete experimental procedure more clearly. There are some suggestions already placed in the manuscript (the pdf with the comments is attached).

Reviewer 2 Report

The manuscript of Wei Pan et al. “Experimental study on enhancing microbial desulfurization of sulfide ores by ultrasonic” provide interesting data of a novel methods for microbial desulfurization of sulfide ores. The effect of the particle size, the ultrasonic action time, and the  ultrasonic power are investigated.

  • The authors show highest efficiency and rate microbial desulfurization by ultrasonic methods in comparison with traditional approach. They confirmed previous result of negative correlation between the particle size of sulfide ore and desulfurization on example of your experiments.

The general results displayed in the paper have  important theoretical and practical significance for the sustainability of the minerals I think, this is a well written, illustrated and insightful study that provides some hypothethis about origin of the deposit.

However, I have a few critical remarks, which I hope, will improve the content of the nice manuscript.

  • The mineralogical description of sulfide varieties with genetic interpretation could be sufficient for comparison with other types of the ores.
  • The paper needs reflected light microphotos of sulfide ore structures studied herein.

I think, the paper is useful  for applied mineralogy and technology research ores and provides criteria to improve desulfurization of sulfide ores. The paper could be accepted with minor revision.
